# Breastfeeding support through wet nursing during nutritional emergency: A cross sectional study from Rohingya refugee camps in Bangladesh

Faria Azad[1], M. A. Rifat[2¤]*, Mohammad Zahidul Manir[3], Nushrat Alam Biva[4]

1 Programme Assistant, World Food Programme of the United Nations, Cox's Bazar, Chittagong, Bangladesh, 2 Consultant, United Nations Children's Fund, Cox's Bazar, Chittagong, Bangladesh, 3 Nutrition Officer, United Nations Children's Fund, Cox's Bazar, Chittagong, Bangladesh, 4 Institute of Nutrition and Food Science, University of Dhaka, Dhaka, Bangladesh

¤ Current address: Directorate General of Food, Government of the People's Republic of Bangladesh, Dhaka, Bangladesh
* rifatahmed011@gmail.com

## Abstract

### Background/Objectives

This study examined the best practices with regard to infant and young child feeding in emergency (IYCF-E) program. This was done by observing a breastfeeding support scenario through wet nursing in Rohingya refugee camps in Cox's Bazar, Bangladesh.

### Methods

Information on demographics, IYCF-E knowledge, wet nursing support, type of constraints faced, and possible ways to overcome such constraints was collected through face-to-face interviews with 24 conveniently selected wet nurses. Linear regression was used to analyze the associations.

### Results

Mean age of wet nurses was 21.6 years; 16.67% had adequate knowledge about IYCF-E; and 29.17% had prior knowledge about wet nursing. Mean age of supported infants was 1.29 months, and 58.33% had a familial relationship with the wet nurses. Duration of breastfeeding support was significantly associated with the wet nurse's age, age of the wet nurses' youngest children, familial relationship with infants, knowledge about IYCF-E, and follow-ups from community nutrition workers (Ps <0.05). The status of facing problems (58.33%) was negatively correlated with duration of wet nursing, although this association was not statistically significant. The most extensively reported problems were as follows: misunderstandings with the infant's family (85.71%), family workload and time limitations (21.43%), household distance (42.86%), and family members' poor compliance (21.43%). Counseling from community nutrition workers (64.29%) and mediation by community leaders (57.14%) played key roles in mitigating such problems. Self-satisfaction (37.50%), counseling

**Data Availability Statement:** All relevant data are within the paper and its Supporting Information files.

**Funding:** The authors received no specific funding for this work.

**Competing interests:** The authors have declared that no competing interests exist.

(62.50%), and religious inspiration (58.33%) were key motivators behind dedicated breastfeeding support.

## Conclusion

Wet nursing in the Rohingya refugee camps in Cox's Bazaar, Bangladesh, was associated with several factors involving both supply and demand. The present findings may help design better IYCF-E programs in similar context.

## Introduction

Wet nursing refers to a surrogate, other than an infant's mother, providing breastfeeding nutrition [1]. As an alternative to a mother's breast milk, wet nursing is considered safer than other modes of infant feeding, such as those involving the use of infant formulas and bottles, and was a common practice before the invention of other alternatives [2].The main purpose of wet nursing is to ensure that an infant is provided with breast milk; this is important for promoting growth and health throughout the life cycle [3]. Breastfeeding can improve infants' resistance to potentially fatal exposures. Exclusive breastfeeding of infants up to six months, and continuing for up to one year, is associated with a reduced mortality rate of 13% before the age of 5 [4]. However, it is crucial to ensure correct and appropriate breastfeeding, which may prove difficult within emergency situations, including life in a refugee camp where fatal exposures due to poor hygiene and sanitation can be problematic [5–7]. Infant and young child feeding in emergency (IYCF-E) program focuses mainly on breastfeeding and prevention of inappropriate use of feeding products. In the context of refugee camps, recent evidence has shown that infant formula donations and distribution has been provided, including during the refugee crisis in Lebanon [8]. Difficulties faced during implementation of IYCF-E programs include resource mobilization, monitoring, coordination, minimizing donations from formula, and improving general awareness so as to translate policy into practice [9, 10].

Global policy on infant feeding is based on respect, protection, and human rights principles. Infants are the most vulnerable potential victims in an emergency. Interrupted breastfeeding heightens the risk of malnutrition, morbidity, and mortality. One challenge in promoting breastfeeding is the mother-to-child transmission of HIV, both among HIV affected and unaffected populations. However, during complex emergencies often characterized by population displacement, food insecurity, and armed conflicts, the risks posed by HIV transmission is often disregarded. Here, breastfeeding is prioritized [11, 12]. Further, if an infant is unable to be breastfed by a biological mother, the viability of re-lactation among wet nurses and donor human milk is subsequently preferred [13, 14].

The current study focused on vulnerable infant breastfeeding support provided through wet nursing within the Rohingya refugee camps in Cox's Bazar, Bangladesh. In the second half of 2017, approximately 0.7 million Rohingya refugees crossed the border from Myanmar to Bangladesh in order to escape death, forced labor, land confiscation, religious intolerance, rape, and other forms of persecution by the Myanmar military regime [15–18]. With a total of 0.25 million pre-resided refugees since 1992, this new influx continued until 2018, leading to a total of 1.2 million Rohingya refugees residing in geographically demarcated registered camps (RCs) and makeshift settlements (MSs) in Cox's Bazar. An extreme humanitarian crisis was declared with respect to nutrition, health, and sanitation [19, 20]. Many infants were unable to receive breast milk owing to either the demise of their biological mothers or their mother's

inability to breastfeed owing to extreme trauma, stress, or other problems [21]. Women who were raped in Myanmar later gave birth to infants whom they might not have wanted. These newborns were left at health facilities and were fortunately adopted by other families [22]. These vulnerable infants required wet nursing, which was managed by humanitarian organizations working with IYCF-E. This study was conducted to examine the overall breastfeeding support scenario within these refugee camps in order to replicate "good practices" in IYCF-E programs.

## Methodology

### Study setting

Wet nurses included in this study were from the Rohingya refugee community and residing in geographically demarcated RCs and MSs. As of the date of this study, there were 2 RCs and 27 MSs located in Ukhiya and Teknaf upazila (sub-district), Bangladesh (Fig 1). Previously arrived refugees had been residing in RCs, and newly arrived refugees were settled in MSs, although there were some areas with both refugees and indigenous residents. The camps were settled in hillside areas with a shortage of drinking water and the possibility of landslides and flashfloods during the monsoon season. The Rohingya have no official written language, and their spoken tongue is similar to that of the locals of Cox's Bazar. Of the total refugees, more than two-thirds were women and children. Rohingya society is highly influenced by their community and religious leaders. The camps were overcrowded, and basic supplies and facilities were below UNHCR standards. Child malnutrition was high within the camps, with a global acute malnutrition rate of 24% [18]. To date, there were 58 nutrition centers providing nutrition services in the camps.

### Study design and sampling

A cross-sectional design was employed in this study. Respondents were selected using convenience sampling from Rohingya refugee camps in Cox's Bazar, Bangladesh. A total of 24 wet nurses were included in the study, and the overall study period extended from August 2018 to January 2019.

### Inclusion criteria and data collection

Wet nurses who provided breastfeeding support for a period of at least two months and provided ethical consent were included in this study. Respondents were contacted through the assistance of community health and nutrition workers (CHNWs) and Rohingya community leaders. Preliminary data were collected through face-to-face interviews, which were conducted at the respondents' tent shelters, after receiving written consent. For further analysis, information was recorded on a questionnaire. The questionnaire was pre-tested and validated through demo interviews and in-house revisions before formal data collection. While interviewing, community nutrition workers and volunteers provided support for communicating and interpreting the Rohingya language. Before interviewing, CHNWs were given clear explanations regarding the questionnaire and aims and objectives of the study so that they could appropriately work with the respondents. Interviews were conducted from September to December 2018. A total of 27 wet nurses were contacted, of whom 24 met the inclusion criteria. The sample size was small because there were very few wet nurses available in the camps. Moreover, the study was designed to observe most effective practices rather than focus on a robust population survey. Therefore, the sample size is considered sufficient.

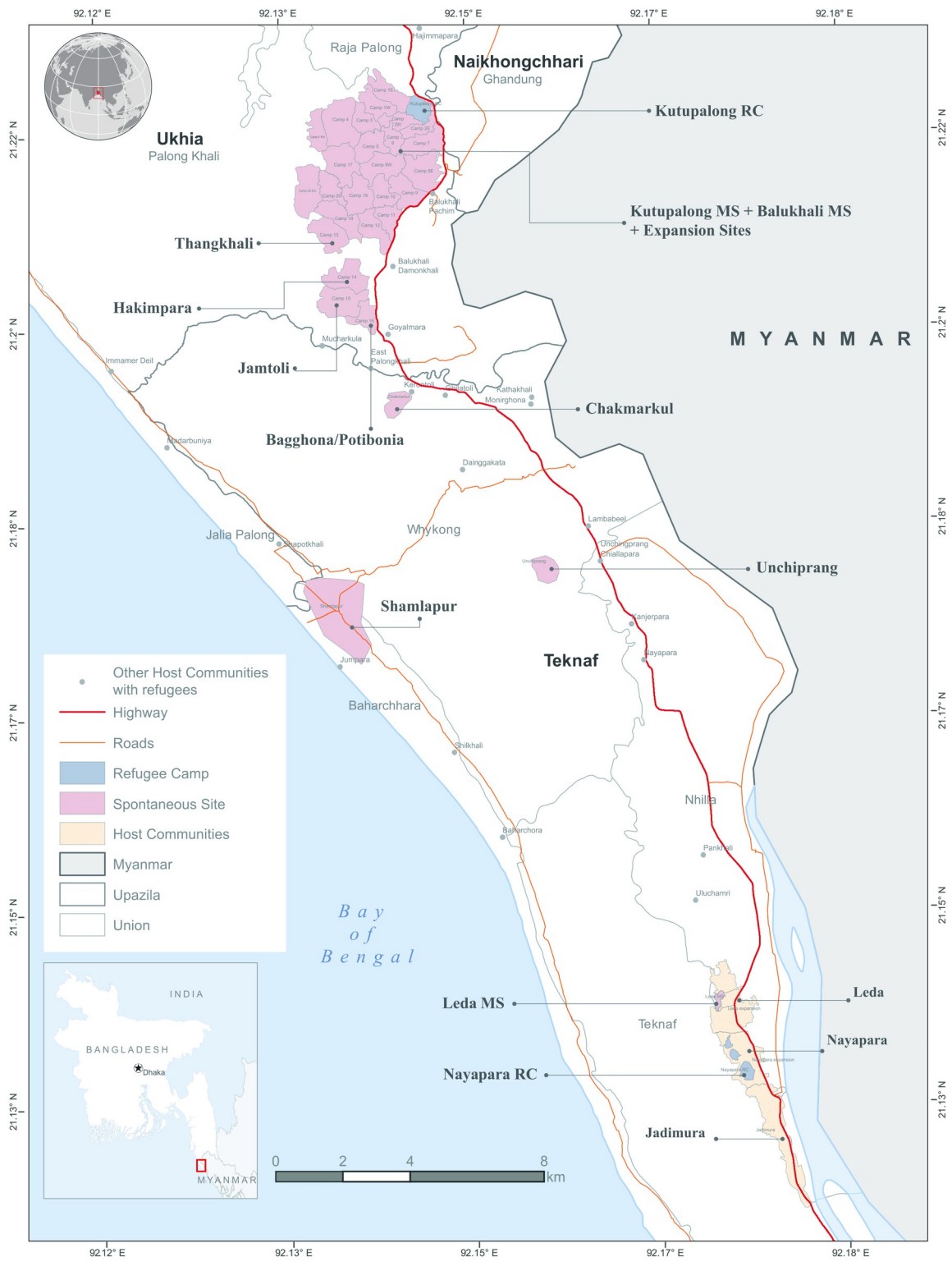

**Fig 1. Rohingya refugee camps map (updated on August 2018).**

## Anthropometric measurement

Respondents' nutritional status was calculated by measuring mid upper-arm circumference (MUAC) and body mass index [23, 24]. MUAC tape, digital bath scale, and height measuring tape were used to measure MUAC, weight, and height, respectively. MUAC, body weight, and height were measured to the nearest values of ±1mm, ±100g, and ±0.1cm, respectively.

## Data analyses

Data quality checks were performed each day through in-house assessments after data collection. Data were entered in SPSS 21.0 (SPSS Inc, Chicago, IL, USA) for analysis. To measure associations between variables, the following linear regression model was used:

$$Y = \beta_0 + \beta X + \varepsilon$$

Wherein, Y = duration of wet nursing, X = independent or predictor variables for Y, $\beta_0$ = regression coefficient for the intercept, β = regression coefficient for the linear effect of X on Y, and ε = random error. All statistical tests were carried out with a 95% confidence interval or α level of 0.05.

## Results

Table 1 outlines all personal information and nutritional statuses from the wet nurses interviewed. Of the respondents, 70.83% (n = 17) were performing wet nursing for the first time, whereas 29.17% (n = 7) were performing wet nursing for the second time. Mean (±standard error) duration of wet nursing was 4.25 ±0.39 months. A total of 16.67%, 29.17%, and 54.17% respondents had good, basic, and poor knowledge, respectively, of IYCF-E. Regarding breastfeeding knowledge, 91.67%, 79.17%, and 20.83% had accurate knowledge about early initiation of breastfeeding, exclusive breastfeeding, and a minimum duration for continuation of

**Table 1. Respondents' personal information and nutritional status.**

| Indicators | Findings (n = 24) |
|---|---|
| **Age** | |
| <18 years | 8.33 (2) |
| ≥18 years | 91.67 (22) |
| Average age (years) | 21.6±0.98 |
| **Family size** | 5.21±0.29 |
| **Number of children** | 2.33±0.19 |
| **Age (months) of the youngest child** | 13.08±1.72 |
| **Education level** | |
| Literate | 4.17 (1) |
| Illiterate | 95.83 (23) |
| **Having husband** | 100 (24) |
| **Education level of husband** | |
| Literate | 16.67 (4) |
| Illiterate | 83.33 (4) |
| **Nutritional status** | |
| Underweight by BMI or BMI for age | 12.5 (3) |
| Undernourished by MUAC (<21cm) | 8.33 (2) |

Data are presented as either % (n), n = number of respondents or mean± standard error

breastfeeding, respectively. Only 29.17% of the respondents had knowledge about wet nursing prior to arrival at the camp. Family (62.5%), community representatives (4.17%), and community nutrition workers (58.33%) were the main sources of wet nurses' knowledge of IYCF-E.

## Information regarding wet nursing

Of the infants that were wet nursed, 45.83% were boys, and 54.17% were girls. Mean (±standard error) age of infants during the initiation of wet nursing was 1.29±0.2 months. The requirement for wet nursing was easily apparent, as 54.17% of infants did not have mothers, and the remaining 45.83% had mothers who were unable to breastfeed. Results revealed that 58.33% of wet nurses had a familial relationship with the infants. In all cases, community health and nutrition workers (CHNWs) first connected wet nurses with infants and they also followed-up the wet nurses mainly on daily (25%) and weekly (75%) basis. The average daily frequency of breastfeeding was 7.08±0.54 times, and each breastfeeding session averaged 25.42 ±1.80 minutes. Few wet nurses (12.5%) reported that they breastfed the infants at night time. The majority (91.97%) of respondents had to receive permission from their husbands or family members before breastfeeding. Every wet nurse was provided non-monetary incentives, such as hygiene kits and baby clothes, by the IYCF-E partners; however, all wet nurses happily agreed to continue their support without any incentive. Most (62.5%) of the wet nurses thought that counseling and follow up from community health and nutrition workers (CHNWs) were key motivators for engagement and continuation with wet nursing. Moreover, self-satisfaction (37.50%), religious motivation (5.33%), and family support (20.83%) were also key factors in this regard.

Most (58.33%) of the respondents also faced difficulties during wet nursing, including misunderstandings with infants' family members, household distance, lack of compliance from their own family members, and household maintenance workloads. Of the wet nurses facing difficulties, 16.67% thought the problems were serious. Community nutrition workers and community leaders played a major role in mitigating these problems. Table 2 represents the types of problems faced by wet nurses and possible ways to overcome them.

Finally, 8.33% wet nurses ceased breastfeeding who were later re-lactated through extensive motivations, counseling and confidence build-up by nutrition professionals. A total of 83.33% respondents expressed that they had positive feelings after performing as wet nurse.

## Associations between variables

Fig 2 shows that wet nursing duration was positively associated with wet nurses' age, age of the wet nurse's youngest child, and frequency of monthly follow-ups from community nutrition workers. Wet nursing duration was longer when wet nurses had a familial relationship with the infants and adequate IYCF-E knowledge. Problems faced were negatively associated with wet nursing duration, although this relationship was not statistically significant. Table 3 shows the statistical relationships between wet nursing duration (months) and other variables analyzed in the linear regression model.

## Discussion

No data were available as to the number of infants without mothers, or the number of mothers unable to breastfeed, in the Rohingya refugee camps. During initial stages of the emergency, refugees had yet to settle and roamed around the camps itinerantly, looking for their relatives and collecting rations. Therefore, the likelihood of a refugee remaining in a particular camp for a long period of time was quite low. Even after settlements were established, refugee migration from one camp to another continued. These factors restricted the number of wet nurses

**Table 2. Constrains faced during wet nursing and ways to overcome.**

| Indicators and levels | Frequency | Percentage |
|---|---|---|
| **Faced any problem while wet nursing (n = 24)** | **14** | **58.33** |
| **Types of problems (n = 14)** | | |
| Poor compliance from own family | 3 | 21.43* |
| Misunderstanding with child's family | 12 | 85.71* |
| Family workload and time limitation | 3 | 21.43* |
| Distance of the child's family | 6 | 42.86* |
| **Severity of the problems (n = 14)** | | |
| Manageable | 2 | 14.29 |
| Often problematic | 8 | 57.14 |
| Serious problematic | 4 | 28.57 |
| **How the problems were solved (n = 14)** | | |
| Counseling by nutrition workers | 9 | 64.29* |
| Mutual discussion with child's family | 6 | 42.86* |
| Support by community leaders | 8 | 57.14* |
| Problems remained unsolved | 1 | 7.14 * |
| **Faced any physical or mental problem (n = 24)** | **6** | **25** |
| **Type of physical or mental problem (n = 6)** | | |
| Physical weakness | 2 | 33.33 |
| Mental stress | 3 | 66.67 |
| Inadequate milk secretion or others | 0 | 0 |

Data are presented as % (n), n = number of samples

*Multiple responses were considered

available to provide support. Additionally, due to cultural practices, the acceptability of wet nursing and/or human milk donations is important issue to be considered within an IYCF-E program [13]. In the present study, we observed that wet nursing was an unfamiliar practice among the Rohingya community. Only 29.17% of the respondents had heard of wet nursing before arriving to the camps. On the other hand, compared to other interventions, wet nursing was less of a focus among implementing organizations owing to a lack of adequate background information. We observed that mothers were sensitive with disclosing information about their child's adoption owing to security issues. All these factors led to fewer available wet nurses despite the large number of children in need of such support. For these reasons, a large sample size was not possible, and respondents were selected conveniently through the assistance of community nutrition workers.

There are few extant studies regarding wet nursing and formal human milk donations both during emergencies and under typical circumstances [13]. However, the little evidence available indicates that wet nursing has been effective and has a positive impact on child growth during a crisis (such as in Ellembelle Nzema, Ghana) [25]. Several factors determine the success of a wet nursing program, including as mother's well-being and motivation, age of the infant, duration of breastfeeding cessation, access to sustained skilled support, one's mental makeup, education, and religious factors [13, 26–27]. Research has shown that even maternal grandmothers could be re-lactated in order to provide breastfeeding support for their grandchildren [28]. We found that 8.33% of respondents, those over the age of 30, were re-lactated after rigorous counseling following post-breastfeeding cessation. The ages of the wet nurses and their youngest child had a significant association with the duration of wet nursing support. Wet nurses with a child between 12 and 24 months provided breastfeeding support for a

longer period than other respondents. One possible reason that mothers with infants less than 12 months old did not provide breastfeeding support to other infants could be the fact that they were breastfeeding their own children.

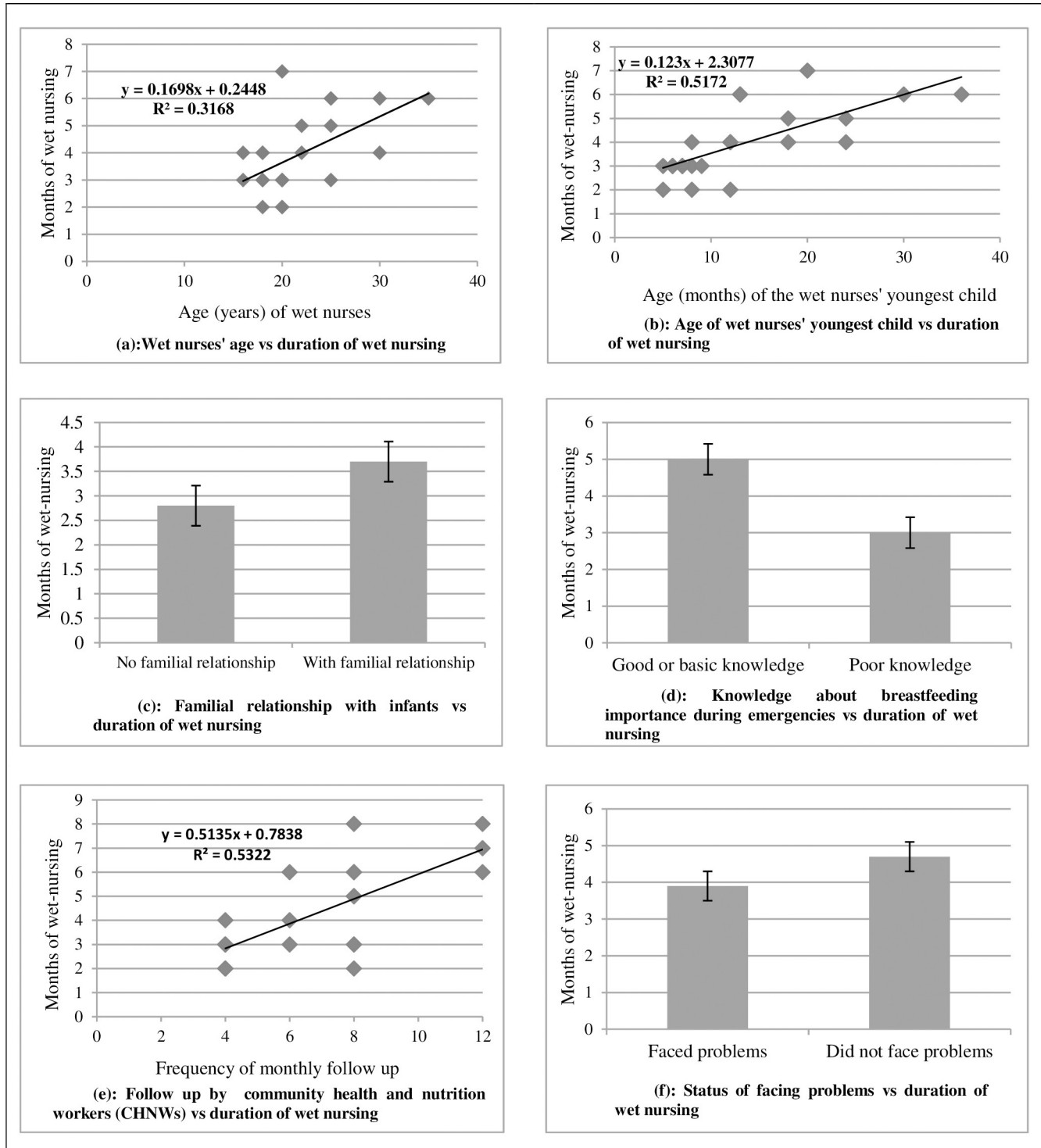

**Fig 2. Relationship between duration of wet nursing (months) and other variables such as wet nurses' age, the age of their youngest child, familial relationship with infants, knowledge about IYCF-E, follow up by community nutrition workers, and problems faced while wet nursing.**

**Table 3. Bivariate regression analysis of duration (months) of wet nursing (Y) with other predictor variables (X).**

| Predictor variables (X) | Coefficient for intercept ($\beta_0$) | Coefficient for X ($\beta$) | $R^2$ | Adjusted R | Standard error | P-value |
|---|---|---|---|---|---|---|
| Age (years) of wet nurses | 0.245 | 0.17 | 0.317 | 0.286 | 1.219 | 0.004 |
| Age (months) of youngest child | 2.308 | 0.123 | 0.517 | 0.495 | 1.025 | 0.000 |
| Familial relationship with infants | 3.1 | 1.4 | 0.239 | 0.204 | 1.286 | 0.015 |
| Knowledge about IYCF-E | 3.0 | 2.0 | 0.498 | 0.475 | 1.044 | 0.000 |
| Frequency of monthly follow up | 0.784 | 0.514 | 0.532 | 0.511 | 1.324 | 0.000 |
| Status of facing any problem | 4.7 | -0.771 | 0.042 | -0.001 | 1.895 | 0.336 |

Tests were carried out at 95% confidence interval

P-values less than 0.05 were considered as statistically significant

We also observed that knowledge regarding IYCF-E, familial relationships with the infants, and follow-ups from community nutrition workers were significantly related to wet nursing duration. We did find that only 16.6% of respondents had adequate knowledge about IYCF-E. One reason behind poor infant feeding could be the high rates of illiteracy within the Rohingya community due to long-term persecution and deprivation in Myanmar [29, 30]. A study in Turkey showed that the possibility of a marriage between "milk siblings," as well as religious concerns, were associated with a problematic wet nursing bottleneck [27]. Apart from religious issues, misunderstandings with a child's family, insufficient support from family members, and family workload were additional significant issues regarding wet nursing support in the present study. The possible reasons behind misunderstanding between wet nurses and infants' family members were mainly security issue of child adoption and future demand of any rendition in lieu of breastfeeding by wet nurses. When dealing with a refugee population, proper understanding of the socio-cultural and economic context aids the process [31]. The Rohingya community is highly influenced by religious beliefs and community leaders, locally referred to as *Majhi*. Religious motivation-based counseling and support from community leaders played a positive role in wet nursing support throughout the present study.

Before migrating to Bangladesh, Rohingya residents were struggling with an acute malnutrition rate that was 50% higher than non-Rohingya residents in Rakhine, Myanmar. Furthermore, the diarrheal-illness rate was five times greater than the general Myanmar population [32]. In the present study, 12.5% of wet nurses were undernourished based on BMI metrics; however, nutritional status was not significantly associated with breastfeeding support, and none of the respondents reported undernourishment as a reason for inadequate breast milk production [33]. Rohingya infants may be more likely to become malnourished because of high exposure to aggravating factors such as disease, poor sanitation, and improper feeding. Studies have indicated that there are high rates of diarrhea, acute respiratory infections, global acute malnutrition, anemia, and poor infant feeding practices in these refugee camps [18, 34]. Thus, collaborative approaches within the health and nutrition sector are likely necessary in order to bolster wet nursing support during nutritional emergencies [6].

## Limitations and strengths

A few study limitations should be noted. As previously mentioned, the present sample size was rather small owing to the unique circumstances influencing our recruitment source. For instance, parents who adopted abandoned babies may have been reluctant to disclose information owing to security issues. Thus, future work will need to improve sampling strategies in order to go beyond the descriptive nature of breastfeeding support during nutritional

emergencies and address ways for improving the health and well-being of vulnerable infants and children.

A few study strengths must also be mentioned. For one, we found evidence to support the idea that wet nursing or formal human milk donations might be feasible, even during a complex emergency, through the dedicated support of implementing organizations. The present findings also support the notion that mothers who cease breastfeeding can be re-lactated through proper counseling and motivation. Finally, best practices for improving breastfeeding support could be replicated in future IYCF-E programs within similar contexts.

## Conclusion

Factors from both the supply and demand side, such as wet nurses' age, age of the wet nurse's youngest child, knowledge of IYCF-E, relationship with supported infants, monthly follow-ups from community nutrition workers, socio-cultural and religious contexts, and compliance from family members and community leaders are crucial when addressing wet nursing support in Rohingya refugee context. Building confidence and counseling are vital for providing a positive psychological experience for women willing to offer such services. There are bottlenecks, such as misunderstandings between families, poor compliance among families, and a lack of knowledge that could be overcome through increased motivation and community support. Thus, effective programs should take these factors into consideration.

## Supporting information

**S1 Questionnaire. Questionnaire in English version.**
(PDF)

**S2 Questionnaire. Questionnaire in Bangla version.**
(PDF)

**S1 Dataset. Minimal underlying dataset.**
(XLSX)

## Acknowledgments

The authors would like to thank Society for Health Extension and Development (SHED) and Social Assistance and Rehabilitation for the Physically Vulnerable (SARPV) for their support during data collection at field level.

## Author Contributions

**Conceptualization:** Faria Azad, M. A. Rifat.

**Data curation:** Faria Azad, M. A. Rifat.

**Formal analysis:** Faria Azad, M. A. Rifat, Mohammad Zahidul Manir.

**Investigation:** Faria Azad, M. A. Rifat, Mohammad Zahidul Manir.

**Methodology:** Faria Azad, M. A. Rifat, Mohammad Zahidul Manir.

**Project administration:** Faria Azad, M. A. Rifat, Nushrat Alam Biva.

**Resources:** Mohammad Zahidul Manir.

**Software:** M. A. Rifat, Nushrat Alam Biva.

**Supervision:** Faria Azad, Mohammad Zahidul Manir.

**Validation:** Faria Azad, M. A. Rifat, Mohammad Zahidul Manir.

**Visualization:** Faria Azad, M. A. Rifat, Mohammad Zahidul Manir, Nushrat Alam Biva.

**Writing – original draft:** Faria Azad, M. A. Rifat, Nushrat Alam Biva.

**Writing – review & editing:** Faria Azad, M. A. Rifat, Mohammad Zahidul Manir, Nushrat Alam Biva.

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
