## [Decision Letter · Decision Letter 0]

2 Aug 2019

PONE-D-19-16700

Breastfeeding support through wet nursing during nutritional emergency in Rohingya refugee camps in Bangladesh: a cross sectional study

PLOS ONE

Dear Dr. Rifat,

Thank you for submitting your manuscript to PLOS ONE. After careful consideration, we feel that it has merit but does not fully meet PLOS ONE’s publication criteria as it currently stands. Therefore, we invite you to submit a revised version of the manuscript that addresses the points raised during the review process.

We would appreciate receiving your revised manuscript by 14 August 2019. To enhance the reproducibility of your results, we recommend that if applicable you deposit your laboratory protocols in protocols.io, where a protocol can be assigned its own identifier (DOI) such that it can be cited independently in the future. For instructions see: http://journals.plos.org/plosone/s/submission-guidelines#loc-laboratory-protocols

We look forward to receiving your revised manuscript.

Kind regards,

Russell Kabir, PhD

Academic Editor

PLOS ONE

Journal Requirements:

1. Please include additional information regarding the survey or questionnaire used in the study and ensure that you have provided sufficient details that others could replicate the analyses. For instance, if you developed a questionnaire as part of this study and it is not under a copyright more restrictive than CC-BY, please include a copy, in both the original language and English, as Supporting Information. Moreover, please include more details on how the questionnaire was pre-tested and validated.

3. In your Methods section, please provide additional information about the participant recruitment method and the demographic details of your participants. Please ensure you have provided sufficient details to replicate the analyses such as: a) the recruitment date range (month and year), b) a description of any inclusion/exclusion criteria that were applied to participant recruitment, c) a table of relevant demographic details, d) sample size justification and a statement as to whether your sample can be considered representative of a larger population, e) a description of how participants were recruited, and f) descriptions of where participants were recruited and where the research took place.

5. Please include your tables as part of your main manuscript and remove the individual files. Please note that supplementary tables (should remain/ be uploaded) as separate "supporting information" files

Reviewers' comments:

Reviewer's Responses to Questions

**Comments to the Author**

1. Is the manuscript technically sound, and do the data support the conclusions?

Reviewer #1: Partly

Reviewer #2: Yes

Reviewer #3: Partly

2. Has the statistical analysis been performed appropriately and rigorously? 

Reviewer #1: Yes

Reviewer #2: No

Reviewer #3: Yes

3. Have the authors made all data underlying the findings in their manuscript fully available?

Reviewer #1: Yes

Reviewer #2: Yes

Reviewer #3: Yes

4. Is the manuscript presented in an intelligible fashion and written in standard English?

Reviewer #1: Yes

Reviewer #2: Yes

Reviewer #3: Yes

5. Review Comments to the Author

Reviewer #1: This is a very interesting research work, written concisely.

Still, a few points are raised for better understanding of the reader.

Use of "in" twice in the title

Use of cross sectional study in the short title

Language grammar and punctuations

Use of keywords which are part of the title

The second sentence in the objective "The ultimate....." may be avoided or inserted in the conclusions

An explanation of prevalence of wet nursing in the refugee camp may be narrated before objectives

A sample size of 24. How can it be sufficient for such a quantitative analysis. Justify using literature and theories.

Association from linear regression requires an explanation.

mention significance at measures of association

Reviewer #2: table 1: it would be better if "n" should be mentioned

Table 2 : it would be better if 95% CI is mentioned

in page no. 5 (Discussion), it is mentioned that wet nurses having 12-24'm' baby were able to breast fed for longer period. it would be better if it could be written as 'months' otherwise leading to confusion of 'meter' distance.

Table 3: 2nd line : age (m) of youngest 'children' could be "child'

Reviewer #3: The study is innovative and relevant especially with current refugee situations globally.

There are a few typographical errors and sentence construction should be reviewed generally.

Further clarification is needed on how a sample size of 24 was gotten.

The authors should review some more literature to make their discussion more engaging.Has wet nursing been done elsewhere?what were the outcomes of such similar studies?

Regarding the nutritional status of the on infant,I am not sure what to make of the information.Was this an attempt to provide information about the prevalence of malnutrition among the infants in that refugee camp or to compare the nutritional status of the infants being wet nursed to those without?

All the best

6. PLOS authors have the option to publish the peer review history of their article (what does this mean?). If published, this will include your full peer review and any attached files.

Reviewer #1: Yes: Dr. Asharaf Abdul Salam

Reviewer #2: Yes: Bhabani Prasad Acharya

Reviewer #3: No

---

## [Author Response · Author response to Decision Letter 0]

31 Aug 2019

Response to editor’s comments:

Thank you for your valuable comments. We have addressed each of your points below:

1. The revised manuscript has been prepared as PLOS ONE's style requirements, including those for file naming.

2. Questionnaire (both in English and Bengali version) used in this study has been provided as “Supporting Information”.

3. Copyediting of manuscript for language usage, spelling, and grammar has been accomplished with the help of professional English editors from Editage (www.editage.com). Thanks for your guidance in this regard. 

4. A copy of the manuscript showing the changes by highlighting them has been uploaded as a “Supporting Information” file. The edited manuscript has been provided as “Manuscript”. 

5. Methods section has been revised accordingly. The demographic and geographic perspective has been addressed. Data collection and data analysis methods have been elaborated for better understanding. Justification of small sample size has been provided. How the questionnaire was pre-tested and validated has been addressed. 

6. Data availability statement has been revised and “Minimal Underlying Dataset” has been provided. 

Response to the comments from Reviewer-1:

Thank you very much for appreciating our work. We highly care your comments thus provided the best effort to address the issues as following: 

1. The title has been revised by professional English editor and has been changed to “Breastfeeding support through wet nursing during nutritional emergencies: a cross sectional study from Rohingya refugee camps in Bangladesh”.

2. The short title has also been revised and has been changed to “Breastfeeding support through wet nursing”.

3. Considering the comment from you as well as academic editor we have made the manuscript revise by professional English editing experts. 

4. There are very few existing literatures regarding wet nursing in refugee context. However, we have focused few examples of wet nursing (such as in Ellembelle Nzema, Ghana and in Turkey) in the discussion section correlating to our findings. In the revised introduction section we mainly focused on the current subject context.

5. Justification of small sample size has been provided in Methods section as well as in Discussion section showing the evidence of available literature. 

6. Associations from linear regression have been explained in Discussion section showing other evidences from literature.

7. Significance level (P) at measures of association has been mentioned.

Response to the comments from Reviewer-2:

Thanks for your valuable comments which helped us to improve the manuscript quality. We have revised the manuscript as per your comments as following: 

1. We have carefully revised the tables, statistical units, and units of measurements in the appropriate places. 

2. In table 1, “n” has been mentioned. 95% CI has been mentioned in Table 3. To mention the age, the term “month” has been used instead of “m”.

3. In tables and figures, terminologies and grammatical issues have been corrected accordingly. 

Response to the comments from Reviewer-3:

We are glad to hear the appreciation of our work from you. Thank for adding value to our work through your valuable comments. However, this is how we considered your comments and revised the manuscript: 

1. The manuscript has been edited by professional English editors. 

2. Justification of small sample size has been discussed in methodology and discussion section which are highlighted. 

3. Some recent evidence of wet nursing has been cited in discussion section. The discussion section has been revised with more evidences from literature.

4. Under-nutrition prevalence in the camp was provided to portray the aggravating factors in the camps. However, some part of this discussion has been replaced to the methodology section (study setting) for better understanding. 

We highly appreciate your comments and look forward to hearing from you regarding the revised manuscript. Thank you from our team.

---

## [Editor Report · Decision Letter 1]

12 Sep 2019

Breastfeeding support through wet nursing during nutritional emergency: A cross sectional study from Rohingya refugee camps in Bangladesh

PONE-D-19-16700R1

Dear Dr. Rifat,

We are pleased to inform you that your manuscript has been judged scientifically suitable for publication and will be formally accepted for publication once it complies with all outstanding technical requirements.

With kind regards,

Russell Kabir, PhD

Academic Editor

PLOS ONE
---

## [Editor Report · Acceptance letter]

24 Sep 2019

PONE-D-19-16700R1 

Breastfeeding support through wet nursing during nutritional emergency: A cross sectional study from Rohingya refugee camps in Bangladesh 

Dear Dr. Rifat:

I am pleased to inform you that your manuscript has been deemed suitable for publication in PLOS ONE. Congratulations! Your manuscript is now with our production department. 

With kind regards,

on behalf of

Dr. Russell Kabir 

Academic Editor

PLOS ONE